# Sustainability Score Comparison of Welding Strategies for the Manufacturing of Electric Transportation Components

**Elizabeth Hoyos** [1,*], **María Camila Serna** [1], **Jeroen De Backer** [2] and **Jonathan Martin** [2]

1   Department of Mechanical Engineering, Universidad EIA, Envigado 055428, Colombia
2   Friction Welding and Processing Section, TWI Ltd., Cambridge CB21 6AL, UK
*   Correspondence: elizabeth.hoyos@eia.edu.co

**Abstract:** Sustainability scores can be used to assess manufacturing strategies, going one step beyond a standard economic assessment. This work uses a previously proposed methodology to evaluate two of the most common welding processes for aluminium alloys that are specifically used in the fabrication of components for the transport industry based on their advantages in generating lightweight and dimensionally efficient parts. For comparison and as proof of concept, two welding methods were selected: Friction Stir Welding (FSW) and Gas Tungsten Arc Welding (GTAW). FSW attained a higher overall sustainability score. Values were calculated for an existing aluminium product, which was part of the opening and closing system of an electric train door, and the final score was 0.78 from FSW compared to 0.69 from GTAW, which was 11% higher in FSW compared to the conventional arc welding process. The analysis carried out included economic, physical, social, and environmental impacts. Finally, an example pertinent to a current EV component is described and considered along with a plan to determine the best welding process for a particular application, and with the calculations, the score obtained for GTAW was 0.43 and 0.68 for FSW, which was 36% higher that the result for the conventional arc welding process.

**Keywords:** sustainability; electric vehicles; welding; railway system





## 1. Introduction

The main goal of the article is to present a comparison of different welding processes in terms of sustainability. The work is focused on two different applications, electric car battery trays (EV) and a component of the door of an electric train, and both results are presented quantitatively in order to show which process is the best in terms of each indicator.

Since the early 2000s, 'sustainability' has become a trending term due to the growing demand to care for the environment and its resources. Currently, in Europe, 25% of the continent's energy consumption corresponds to the industrial sector [1]. This percentage is considered important and shows the necessity of implementing sustainable techniques to reduce the resources required; thus, the world is moving towards green manufacturing, since it is a way to optimize consumption. Most commonly, the selection of a manufacturing process is based on three indicators: quality, speed, and cost. Having a standardized indicator for sustainability may allow the industry sector to compare alternatives in light of this approximation instead of a purely economic one, facilitating the selection of more sustainable alternatives. It is important to highlight that, in the future, sustainable manufacturing aims to create different manufacturing techniques with the lowest impacts possible [2], and the companies must adapt themselves to the new challenges presented to stay competitive [3]. Although different approaches have been assessed, the development of a sustainability score which is generally accepted continues to be a complex task for the sector.

Accurate sustainability assessments are complex and typically combine multiple research areas. Life Cycle Assessment (LCA) is the most common method used to assess

the environmental impacts associated with a product or service throughout its entire life cycle, from extraction of raw materials to end-of-life disposal. LCA can be used to compare the environmental impacts of different products or services. However, in order to compare the economic, physical, social, and environmental impacts of different options, multi-criteria decision analysis (MCDA) is required, which is a decision-making technique that allows decision makers to consider multiple criteria or factors when evaluating alternatives. This approach has been proposed within the scope of this work by means of a weighting function [4]. Finally, through the integrated reporting framework, financial and non-financial information can be combined to provide a more comprehensive view of an organization's sustainability performance.

The term sustainability can be addressed in different areas, and the article called Integrating Green Lean Six Sigma and Industry 4.0 provides a conceptual framework written by Kaswan et al. to show sustainability enhancement using a GLSS-Industry 4.0 [5]. Furthermore, the book Sustainability, written by Kent E. Portney, breaks down the term based on energy, business, communities, cities, and consumption, with the last category introducing different areas, such as materials and production [6]. The use of green technologies has been increasing due to the need to reduce the waste generated during production. Authors such as Tamang et al. define manufacturing sustainability as a method to produce components by achieving overall efficiency in terms of the three E's: Economic, Environmental, and Equity/social aspects, these three pillars (view Figure 1) are essential to produce components responsibly [7–10]. The economic and environmental pillars involve a common term, energy consumption. In manufacturing, this is the primary resource of energy and carbon footprint generation in the industry, and there is a necessity to reduce that footprint, although the reduction of these indicators is a hard task to accomplish. For example, the efficiency of machine tools is less than 30% [11], which shows the necessity to improve the technology to enable manufacture with energy consumption as low as possible, and designing this type of machinery with higher efficiencies requires investment in technology and capital [12].

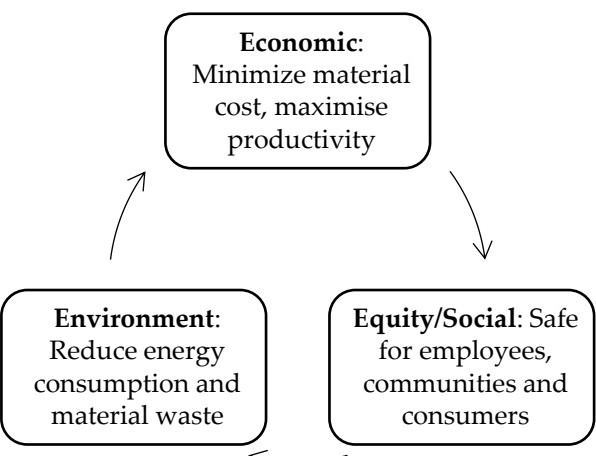

**Figure 1.** Pillars of sustainability (based on [10]).

Given the rise of sustainability and its potential use as a design criterion in manufacturing, some techniques have been introduced. One book outlines innovations in manufacturing for sustainability [13], and enumerates several applications researched, such as sustainability in welding and processing, dry and near-dry machining techniques for 'green' manufacturing, and a research framework of sustainability in additive manufacturing. The main goal of this article is to use one of these strategies to perform a comparison, as a validation exercise, of the calculated sustainability scores for two different welding processes on a given industry problem, as presented below.

## 2. Preliminary Concepts and Proposed Work

Below are the basic and necessary concepts to address the selected case study, such as the material used, the processes analysed, and a description of the train component used for this exercise.

### 2.1. Aluminium Welding

Aluminium alloys are selected for applications in several industries, particularly those related to transportation, such as the manufacturing of ships, trains, and airplanes, among others, all based on the strength-to-weight ratio and corrosion resistance of this family of alloys. Although these materials have advantages, their use is avoided in many cases because of the difficulties presented in fusion welding due to susceptibilities to hydrogen embrittlement, porosity formation, solidification cracking, and distortion after welding, which are all based on the high thermal conductivity and the low melting point of aluminium alloys [14]. The oxide film and other organic impurities presented on the surface of aluminium can also increase the probability of producing defects when welding [15].

Furthermore, the production of aluminium is increasing due to the fact that it is considered a highly recyclable material that can be reused multiple times without losing its properties (view Figure 2, the graph contains the data of the total aluminium produced around the world according to the International Aluminium Institute [16]), and aluminium recycling reduces 94% of the carbon footprint compared to producing the metal from primary aluminium. Indeed, the energy demand and carbon footprint of recycled aluminium has been reduced by 49% and 60%, respectively, since 1991 [17].

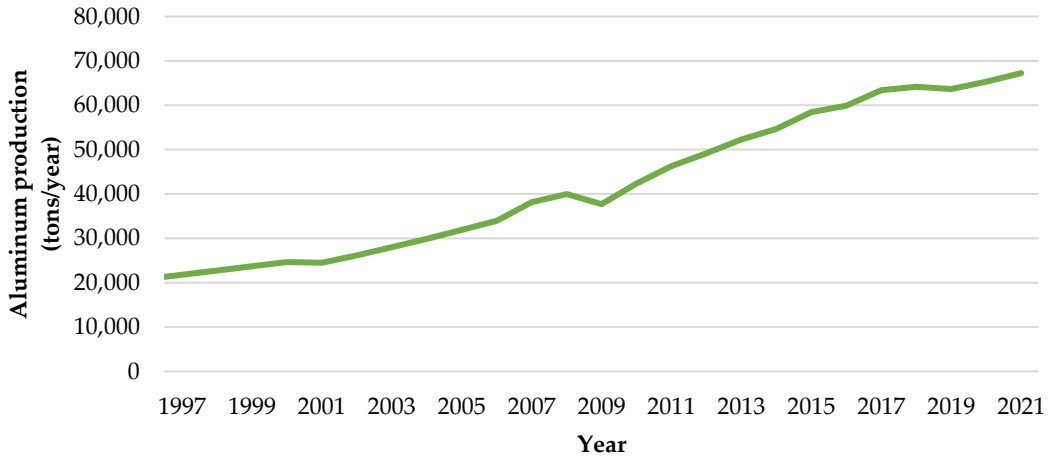

**Figure 2.** Aluminium production (based on [16]).

### 2.2. Selection of a Welding Process

For the welding of aluminium alloys, the conventional method is Gas Tungsten Arc Welding (GTAW), a fusion process that uses an arc formed between a non-consumable tungsten electrode and the workpiece as heat source [18] (Figure 3a), this process uses as a protection an inert gas such as argon, helium, or a mixture. Execute this method will be a challenge, for instance, it can produce hot cracking and softening in the weld fusion zone and HAZ, this can be responsible for decrease of mechanical properties [8]. Due to the aforementioned challenges, alternative joining processes have been developed, one of which is Friction Stir Welding (FSW), invented and patented in 1991 at TWI. FSW is a solid-state welding process that uses a non-consumable tool that rotates and travels along the workpiece [19] (view Figure 3b). The movement during the welding process produces heat through friction, mixing the softened material to produce the weld [1]. The tool has two main parts, shoulder and pin, both playing a crucial role in the welding process. The first is responsible for the heat generation and applies a downward forging force [20], and because the function of the pin is to transport the plasticized material along the joint [21], it

can be designed with different geometries. In literature different tool types can be found, the pin can be cylindrical, conical, and threaded, and, moreover, the shoulder can have a different outer surface shape, such as concave, convex, scroll, etc. [21]. This process is commonly used to weld aluminium, magnesium, and zinc [22]. The selected welding process basic parameters are (a) GTAW [23] and (b) FSW.

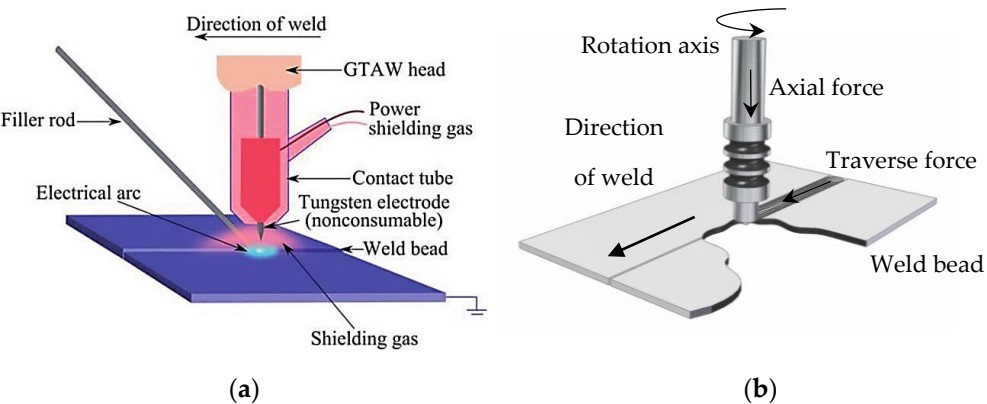

(**a**)            (**b**)

**Figure 3.** Selected welding processes basic parameters: (**a**) GTAW [23] and (**b**) FSW.

Table 1 lists some of the advantages of FSW over GTAW.

**Table 1.** Advantages of FSW over Gas Tungsten Arc Welding (GTAW).

| Environmental | Social | Economical |
|---|---|---|
| Free of consumables, shielding gases, fumes, smoke, or radiation [24] | Zero-emission of smoke or ultraviolet, X-ray, or infrared radiation [25] | Lower energy requirements compared to fusion processes [24] |
| Elimination of the use of solvents [24] | Fully mechanized process, safer for the operator [25] | Increase in productivity [26] |
| | Decrease in the number of problems associated with shielding gases, fumes, smoke, and radiation [27] | |
| | Cardiovascular and lung diseases caused by welding fumes and metal dust from post-weld grinding operations [28] | |
| | Musculoskeletal disorder caused by repetitive manual welding [29] | |
| | Lower accident rates (eyes and skin burn) associated with exposure to intense light and radiation [30] | |

Although FSW offers advantages over GTAW, it can be noted that this process requires specific machinery, and for the execution, four kinds of machines can be used: conventional machine tools, dedicated FSW machines, or custom-built machines and industrial robots. The machines must react the different loads involved during the welding, including axial force, traverse force, side force, and torque [31]. Indeed, the typical machines used in the case of FSW are more expensive and larger than GTAW.

Considering the column of environmental advantages, FSW is considered a green technology as it does not emit any excessive noise, does not cause soil pollution, and does not require the use of solvents during the process [32].

### 2.3. Case Study Description

Metro de Medellín is a Colombian company founded in 1979 whose main activity is to manage and operate the mass transit system of the city of Medellín [33], which includes

trains, trams, buses, and several cable car lines [34]. On the doors of their trains, there are components known as ties, which are located on the upper flange of every door (view Figure 4a) as side-to-side mirror pieces, and which are coupled with other elements to create a mechanism that allows movements both out and away from the centre of the assembly, thereby opening and closing the doors. The tie material is aluminium alloy 6063-T6 (view Table 2), and Figure 4b shows a 3D representation of the part.

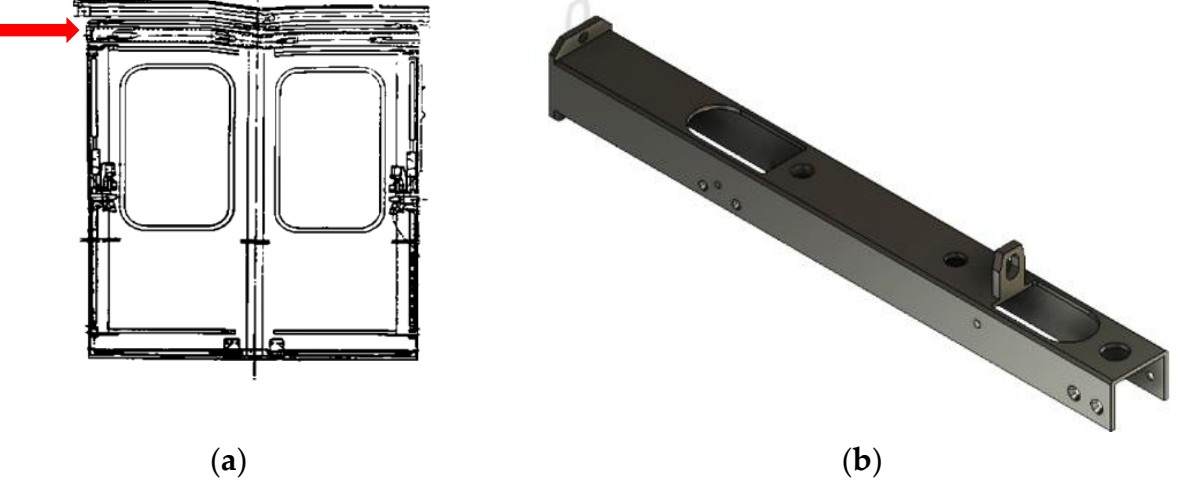

(**a**)          (**b**)

**Figure 4.** Component of the train door: (**a**) Geometry of the tie; (**b**) Tie's location.

**Table 2.** Base material properties (AA6063-T6).

| Properties | Value |
| --- | --- |
| Yield Strength [35] | 214 MPa |
| Toughness [36] | 25.8 J |

The manufacturing of the component described includes multiple steps, such as cutting, grinding, drilling, and welding. For this work, the focus is on welding. The part was made using two different processes: GTAW and FSW. Each of the methods is performed separately. For example, Figure 5 shows the weld locations selected for each process based on its specificities and advantages (Figure 5a includes dotted red lines to show weld locations for GTAW, in Figure 5b can be seen the weld locations watching the blue and red zones). Additionally, considering the different underlying physical principles, the specific parameters of each process vary substantially, and Table 3 shows a summary of the welding parameters used in both cases.

**Table 3.** General welding parameters.

| GTAW | FSW |
| --- | --- |
| Filler material: ER4043 | Rotation Speed: 600 RPM |
| Current: 175 A | Welding Speed: 600 mm/min |
| Voltage: 18 V | Tilt angle: 0° |
| | Shoulder with scroll |
| | Shoulder diameter: 16 mm |
| | Pin diameter: 8 mm |

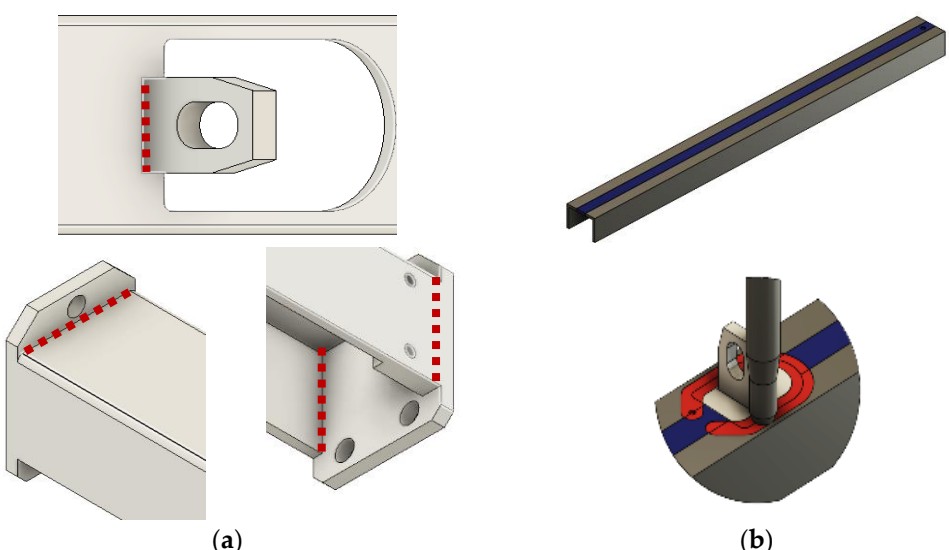

(**a**)                                                                                                   (**b**)

**Figure 5.** Weld locations when using (**a**) GTAW and (**b**) FSW.

### 3. Methodology for Calculating Sustainability Scores

Sustainability can be measured in different forms because different types of work have been carried out, such as social, economic, environmental, efficiency, energy, quality, etc. [2], although the information presented in the article only used a few of them due to the specificity of the applications. Several authors have been carrying out such work, particularly on manufacturing, furthermore, Jamal et al. described a procedure to calculate and compare welding methods with the three aforementioned e's (environmental, economic, and equity impacts). This particular assessment also includes the physical aspect, i.e., strength, considering that in manufacturing, mechanical properties are essential for the performance of the components [37,38]. The methodology of the present work is based on the aforementioned article, and in order to calculate sustainability, Equation (1) was proposed; each term of the equation is presented below, from Equations (2) to (11). In this case, the weights assigned to each impact were taken directly from the original analysis proposed, and the weights were obtained from a survey of different professions, such as engineers, welders, professors, etc., with each one of these professionals having to select for each category what they considered the percentage of incidence [37]. It should be noticed, therefore, that the calculations were made by normalizing each one of factors, and each one of the results correspond to a percentage (all scores are set between 0 and 1). For this reason, higher scores in each category mean a better result in terms of sustainability.

$$Score = Physical\ impact * W + Environmental\ impact * X + Economic\ impact * Y + Social\ impact * Z \qquad (1)$$

where:

$$W = 0.296 = Weight\ associated\ to\ physical\ impact \qquad (2)$$

$$X = 0.24 = Weight\ associated\ to\ environmental\ impact \qquad (3)$$

$$Y = 0.198 = Weight\ associated\ to\ economic\ impact \qquad (4)$$

$$Z = 0.266 = Weight\ associated\ to\ social\ impact \qquad (5)$$

#### 3.1. Physical Impact

The expression contains the yield strength and toughness of the base material (AA 6063-T6). For data on the mechanical properties, applicable standards were used [26].

Furthermore, for the estimation of the yield strength and toughness of the welds, tensile tests were carried out using an INSTRON 3345 universal testing machine.

$$Physical\ impact = \frac{\frac{Weld\ toughness}{Base\ material\ toughness} + \frac{Weld\ yield\ strength}{Base\ material\ yield\ strength}}{2} \tag{6}$$

### 3.2. Environmental Impact

For the calculation of the environmental impact (view Equation (7)), several terms should be considered as follows. The first term (weld emissions) involves the different materials emitted to the surroundings—for example, carbon dioxide ($CO_2$) during the execution of the weld (see Equation (8)). The second one, wastage, refers to the difference between base material, filler metals used, and the mass of the welded component, and lastly, weld mass, which refers to the mass obtained in each welding process. For these last two elements of the environmental impact factor, wastage, and weld mass, an equation is not proposed, and only estimated values for each selected case. Different assumptions have been made for each welding process, all of which are described in Section 4.

$$Environmental\ impact = 1 - \frac{Weld\ emissions + \frac{Wastage}{Weld\ mass}}{2} \tag{7}$$

where:

$$Weld\ emissions = \frac{Metal\ particulate\ ratio + \frac{Carbon\ footprint}{Carbon\ footprint\ limit} + \frac{Auxiliary\ material\ usage}{Auxiliary\ material\ limit}}{3} \tag{8}$$

### 3.3. Economic Impact

This term contains different factors such as labour, consumable and energy costs used during the welding, equipment cost, and welded part cost, with the last one being the individual unwelded section. The expression for calculating the economic impact is found in the equation presented below.

$$Economic\ impact = 1 - \frac{Weld\ time * Labour + Consumable Equipment + Energy consumption * Energy cost}{Welded\ part\ cost} \tag{9}$$

### 3.4. Social Impact

An incident rate describes the number of occurrences in a specific period (see Equation (10)), and this factor is associated with the health and safety of the employees in the performance of work activities [39]. Social impact is calculated with Equation (11). It is important to clarify that Equation (11) was modified from the original source, since it only had the second term.

$$Incident\ rate = \frac{incidents * number\ of\ hours\ work\ in\ a\ year}{Real\ number\ of\ hours\ work} \tag{10}$$

$$Social\ impact = 1 - average\left(\frac{Incident\ rate}{Maximum\ incident\ rate}\right) \tag{11}$$

## 4. Results Analysis

The initial numeral presents in detail the values used for the calculations that were made following the previously described equations for the train component manufactured in Colombia, with the values obtained for each term (see Tables 4 and 5). It must be considered that the calculations were made with the information found, and that some authors report different reviews about sustainability of manufacturing. Gunasekaran et al., for example, reviewed the literature and concluded that the assumptions we have to make to estimate the costs and benefits of sustainable efforts are unrealistic, and that to attain

more realistic data, the academic researchers or practitioners must conduct research in more detail [40]. The following section will raise the use of the sustainability scores for battery trays and the EV industry, using the UK data for impact calculations, where the potential use of this approach is raised to justify the selection of alternative manufacturing processes.

**Table 4.** Summary of sustainability score calculations for GTAW.

| Physical Performance | | Environmental Impact | | Economic Impact | | Social Impact | |
|---|---|---|---|---|---|---|---|
| Weld Yield Strength (MPa) | 60.0 | Weld emissions (kg) | 0.08 | Consumable cost (USD) | 26.05 | Incident rate (Days away from work/100 employees) 2019 | 13.95 |
| Weld Toughness (J) | 1.08 | Auxiliary material usage (g) | 186,666.67 | Labour cost (USD/min) | 0.12 | Maximum incident rate (Days away from work/100 employees) 2019 | 160 |
| | | Material auxiliary limit (g) | 816,666.67 | Weld time (min) | 16.67 | Incident rate (Days away from work/100 employees) 2020 | 105 |
| | | Wastage (g) | 0 | Energy consumption (kW) | 3.15 | Maximum incident rate (Days away from work/100 employees) 2020 | 165 |
| | | Weld mass (g) | 3303.63 | Energy cost (USD/kWh) | 0.05 | Incident rate (Days away from work/100 employees) 2021 | 131.91 |
| | | Carbon footprint limit (kg) | 986.28 | Equipment cost (USD) | 0.03 | Maximum incident rate (Days away from work/100 employees) 2021 | 170 |
| | | Carbon footprint (kg) | 0.14 | Weld part cost (USD) | 246.38 | | |
| | | Metal particulate ratio (non-dimensional) | 0 | | | | |

**Table 5.** Summary of sustainability calculations for FSW.

| Physical Performance | | Environmental Impact | | Economic Impact | | Social Impact | |
|---|---|---|---|---|---|---|---|
| Weld Yield Strength (Mpa) | 102.9 | Weld emissions (kg) | 0 | Consumable cost (USD) | 0.60 | Incident rate (Days away from work/100 employees) 2019 | 6.50 |
| Weld Toughness (J) | 1.86 | Auxiliary material usage (g) | 0 | Labour cost (USD/min) | 0.12 | Maximum incident rate (Days away from work/100 employees) 2019 | 76.49 |

**Table 5.** *Cont.*

| Physical Performance | Environmental Impact | | Economic Impact | | Social Impact | |
|---|---|---|---|---|---|---|
| | Material auxiliary limit (g) | 0 | Weld time (min) | 1.67 | Incident rate (Days away from work/100 employees) 2020 | 6.39 |
| | Wastage (g) | 0 | Energy consumption (kW) | 4.80 | Maximum incident rate (Days away from work/100 employees) 2020 | 36.71 |
| | Weld mass (g) | 3282 | Energy cost (USD/kWh) | 0.07 | Incident rate (Days away from work/100 employees) 2021 | 5.89 |
| | Carbon footprint limit (kg) | 986.28 | Equipment cost (USD) | 0.02 | Maximum incident rate (Days away from work/100 employees) 2021 | 90.99 |
| | Carbon footprint (kg) | 0.03 | Weld part cost (USD) | 246.38 | | |
| | Metal particulate ratio (non-dimensional) | 0 | | | | |

### 4.1. Sustainability Scores as a Tool for the Selection of Manufacturing Strategies

For each welding process, calculations were made to determine the sustainability score [20,21], with Table 4 showing the parameters of GTAW and Table 5 showing the data for FSW [15,16]. Everything related to material and personnel costs was consulted directly with local Colombian suppliers and workshops, for example, and the data collected for engineers and technicians' hourly rates was averaged. For the social impact of each process, different data was located on the database for incident rates in Colombia, and it was necessary to consider FSW as a machining process given the similarities between equipment types and the lack of familiarity of the local metal–mechanical sector with this type of welding process [41].

For the presented case, the material of the component was a 6XXX series aluminium alloy, an alternative with high strength-to-weight ratio as well as considerable thermal and electrical conductivities. FSW commonly presents better mechanical properties compared to GTAW [42], in particular weld yield strength and toughness.

Aspects considered to calculate the environmental impact involve the use of auxiliary material—in this case, the use of Argon as a shielding gas for GTAW—and for FSW no emissions were accounted for. It is worth mentioning that the effect of electrical energy generation and its consumption during welding was considered for both processes. In Colombia, hydropower is the norm, and with no fossil-fuel burning, it is supposed to have a smaller carbon footprint per unit of energy than electricity generated from other sources.

Another factor in the environmental aspect is wastage. In this case, FSW tool wear was considered negligible, since the length of the welds required is close to 1 m, and material loss for aluminium welds in similar conditions in this case study start to be accounted for after approximately 100 m. The wastage was zero in GTAW because the grinding of the piece was not considered.

According to XM, a company from Colombia responsible for wholesale energy market management, every 1 kWh used produces 164.38 g of $CO_2$ [43], and this number was used for the calculation of this term and the carbon footprint limit. The terms associated with the calculation of the weld emissions for FSW are the carbon footprint, carbon footprint limit,

metal particulate ratio, auxiliary material usage, and material usage limit, and according to Equation (9), the result of the wastage is 0. The equipment cost was also considered by finding the average of the prices of different machines used in each process. For example, GTAW used the cost of Dynasty Miller 280, Fronius iwave 190i–230i, among others. Furthermore, in FSW, for this specific case, the search of machinery was focused on milling machines and CNC machines, owing to the fact that these types of machine are present in Colombia, a country that does not have any dedicated machines for executing FSW. The results of the sustainability scores calculated with the previous considerations are summarized in Table 6, and they show a bar chart comparing the results of each term for both welding processes (each value is presented in Figure 6), whilst Figure 7 shows the overall sustainability score.

**Table 6.** Sustainability score for GTAW and FSW.

| Aspect | GTAW | FSW |
|---|---|---|
| Physical performance | 0.24 | 0.39 |
| Environmental Impact | 0.96 | 1.00 |
| Economic Impact | 0.89 | 1.00 |
| Social Impact | 0.80 | 0.89 |
| Sustainability Score | 0.69 | 0.78 |

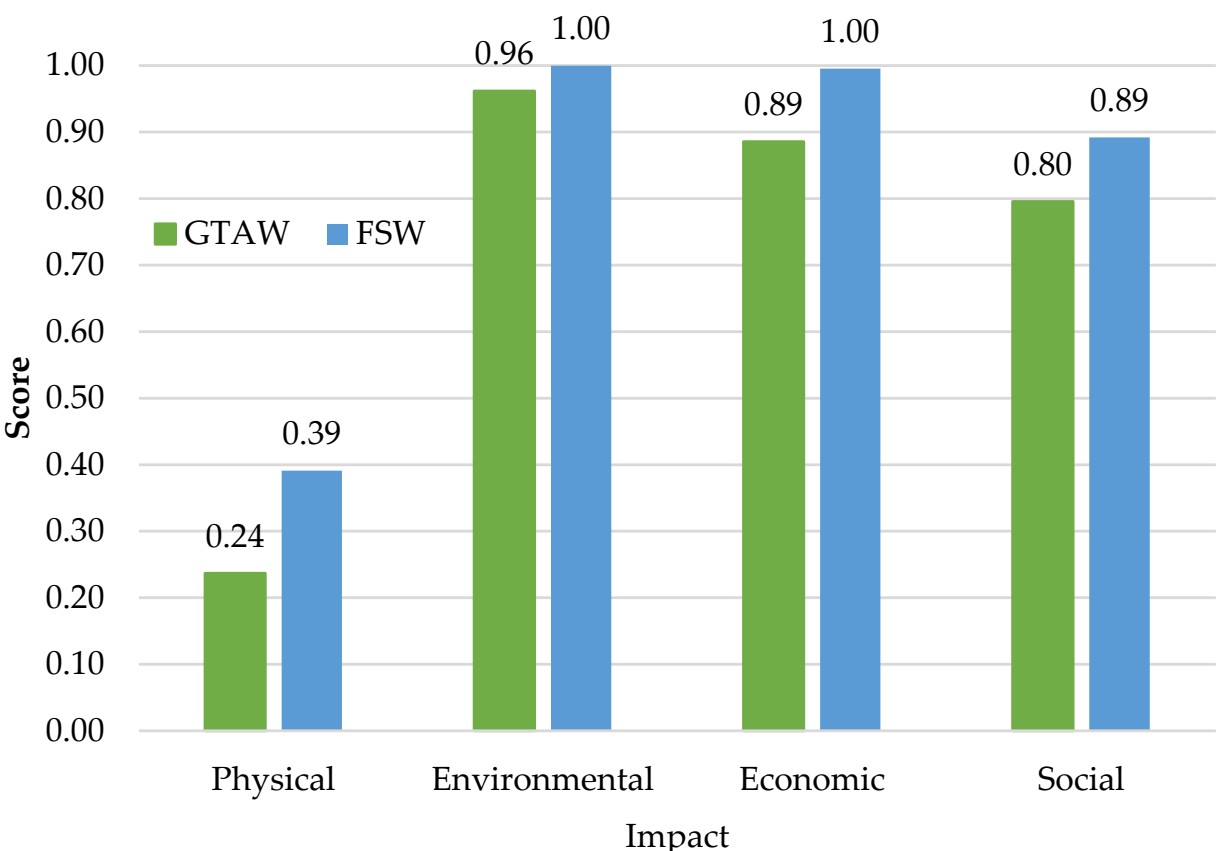

**Figure 6.** Sustainability scores comparison for FSW and GTAW.

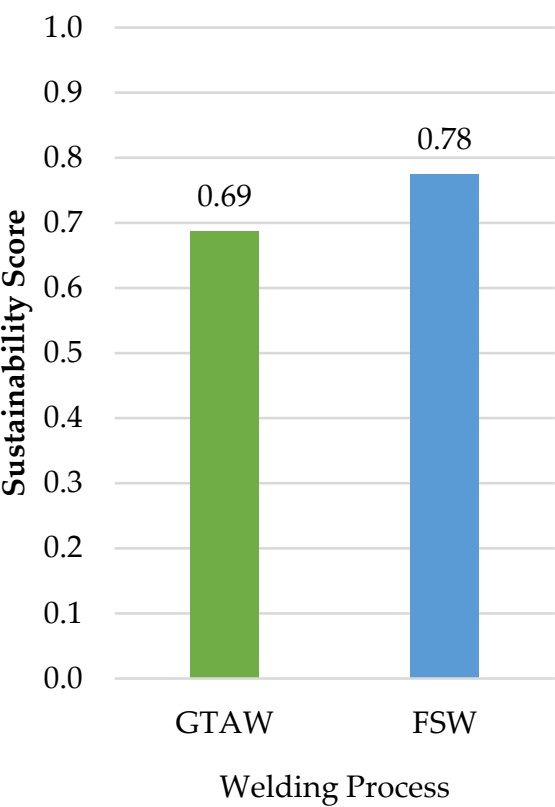

**Figure 7.** Overall score.

### 4.2. Approach to Sustainability Scores in Battery Trays for the EV Industry

EVs are rapidly replacing fossil-fuel cars and the investment in this transition is at a record high. Despite some concerns about high $CO_2$ emissions during the manufacturing of batteries, the overall consensus in academia is that battery–electric vehicles significantly reduce the carbon footprint of a car over its lifetime. Furthermore, EVs entirely eliminate the problem of local air pollution, annually causing tens of thousands of premature deaths globally [44]. The following example of the EV battery tray is described and analysed along with a strategy to determine the most suitable welding process for a particular application.

The function of the battery trays is to contain the batteries cells in crash-resistant, leak-tight, sealed housing. These aluminium structures consist of multiple extruded aluminium profiles that are welded together (double-sided) to form a rigid tray in which the battery modules are subsequently mounted. Aluminium is commonly used for battery trays, and different welding methods have been investigated, including laser beam welding, MIG welding, and FSW. Friction stir welding is commonly used for fabrication of electric vehicle battery trays because this avoids problems of hot cracking, porosity, and element loss [44]. The article titled "High speed friction stir welding of AA6063-T6 alloy in lightweight battery trays for EV industry: Influence of tool rotation speeds", by Patel et al., uses friction stir welding in the assembly of a battery tray. The material used was AA 6063-T6 with 3 mm thickness in a butt configuration. The highest joint efficiency obtained was 72% with a welding speed of 4 m/min and a tool rotation speed of 3500 rpm [45].

For this study, experimental trials were conducted on a FSW Machine at TWI Technology Centre (Yorkshire). It was found that the combined continuous power consumption for a 4 mm thickness SS-FSW in an extruded 6000-series aluminium alloy is 18 kW at 1200 rpm and 500 m/min welding speed. When the rotation speed is increased to 1800 rpm, a welding speed of 1 m/min can be achieved, with an estimated total machine power of 20 kW (during steady state welding). Based on the example battery tray provided by a TWI client, a total of 7 extrusions of 1.5 m length are joined from both sides, resulting in a total weld length of 18 m per tray. Therefore, the total energy input to produce 1 tray is 6 kWh. Due

to the nature of the SSFSW process, no pre- or post-welding operations, such as milling or deburring, are required. A sustainability score comparison was made for battery trays, and the calculations were performed for GTAW and FSW (Tables 7 and 8 show each one). This calculation was assumed to be performed in United Kingdom, and, thus, the real location of the company, the social impact [46], energy cost [47] and labour cost [48] change significantly. The results of the case are summarized in Table 9.

**Table 7.** Summary of sustainability score calculations for GTAW—battery trays.

| Physical Performance | | Environmental Impact | | Economic Impact | | Social Impact | |
|---|---|---|---|---|---|---|---|
| Weld Yield Strength (Mpa) | 60 | Weld emissions (kg) | 0.08 | Consumable cost (USD) | 52.10 | Incident rate (Days away from work/10,000 employees) 2019 | 4536 |
| Weld Toughness (J) | 1.08 | Auxiliary material usage (g) | 3,360,000 | Labour cost (USD/min) | 0.35 | Maximum incident rate (Days away from work/10,000 employees) 2019 | 4536 |
| | | Material auxiliary limit (g) | 14,700,000 | Weld time (min) | 300 | Incident rate (Days away from work/10,000 employees) 2020 | 4158 |
| | | Wastage (g) | 0 | Energy consumption (kW) | 3.15 | Maximum incident rate (Days away from work/10,000 employees) 2020 | 4158 |
| | | Weld mass (g) | 3671.42 | Energy cost (USD/kWh) | 4.99 | Incident rate (Days away from work/10,000 employees) 2021 | 3780 |
| | | Carbon footprint limit (kg) | 986.28 | Equipment cost (USD) | 0.28 | Maximum incident rate (Days away from work/10,000 employees) 2021 | 3780 |
| | | Carbon footprint (kg) | 2.59 | Weld part cost (USD) | 246.38 | | |
| | | Metal particulate ratio (non-dimensional) | 0 | | | | |

**Table 8.** Summary of sustainability calculations for FSW—battery trays.

| Physical Performance | | Environmental Impact | | Economic Impact | | Social Impact | |
|---|---|---|---|---|---|---|---|
| Weld Yield Strength (Mpa) | 102.9 | Weld emissions (kg) | 0 | Consumable cost (USD) | 3.60 | Incident rate (Days away from work/10,000 employees) 2019 | 2086.56 |
| Weld Toughness (J) | 1.86 | Auxiliary material usage (g) | 0 | Labour cost (USD/min) | 0.35 | Maximum incident rate (Days away from work/10,000 employees) 2019 | 4536 |

**Table 8.** *Cont.*

| Physical Performance | Environmental Impact | | Economic Impact | | Social Impact | |
|---|---|---|---|---|---|---|
| | Material auxiliary limit (g) | 0 | Weld time (min) | 18 | Incident rate (Days away from work/10,000 employees) 2020 | 1912.68 |
| | Wastage (g) | 0 | Energy consumption (kW) | 20 | Maximum incident rate (Days away from work/10,000 employees) 2020 | 4158 |
| | Weld mass (g) | 16,200 | Energy cost (USD/kWh) | 1.70 | Incident rate (Days away from work/10,000 employees) 2021 | 1738.80 |
| | Carbon footprint limit (kg) | 11,130 | Equipment cost (USD) | 0.50 | Maximum incident rate (Days away from work/10,000 employees) 2021 | 3780 |
| | Carbon footprint (kg) | 2.23 | Weld part cost (USD) | 246.38 | | |
| | Metal particulate ratio (non-dimensional) | 0 | | | | |

**Table 9.** Sustainability score for GTAW and FSW—battery trays.

| Aspect | GTAW | FSW |
|---|---|---|
| Physical performance | 0.24 | 0.39 |
| Environmental Impact | 0.92 | 1.00 |
| Economic Impact | 0.31 | 0.82 |
| Social Impact | 0.31 | 0.65 |
| Sustainability Score | 0.43 | 0.68 |

For the case presented, the equipment cost is different from the aforementioned case, and, thus, the welding is executed in the United Kingdom, and the prices of the machinery used for FSW were for only CNC and dedicated machines. This is why the search was focused on machineries such as Doosan DNM 750 L II, Manford VH-1300, and Hartford 1570 for CNC machines, and Stirweld, Bond technologies RM7, and others for dedicated machines.

Figures 8 and 9 show the results obtained for the battery tray case, and it can be seen that all the scores for FSW are higher, except for the economic impact, and this is mainly the result of two factors: the cost of electric energy in the United Kingdom, currently at around 34.0 p/kWh (pence per kilowatt hour) for electricity from October 2002 to March 2023 [48], which is higher than electricity prices in Colombia. Regarding the cost of the tool for this specific case, 100 USD compared to the 120 USD for the train tie case.

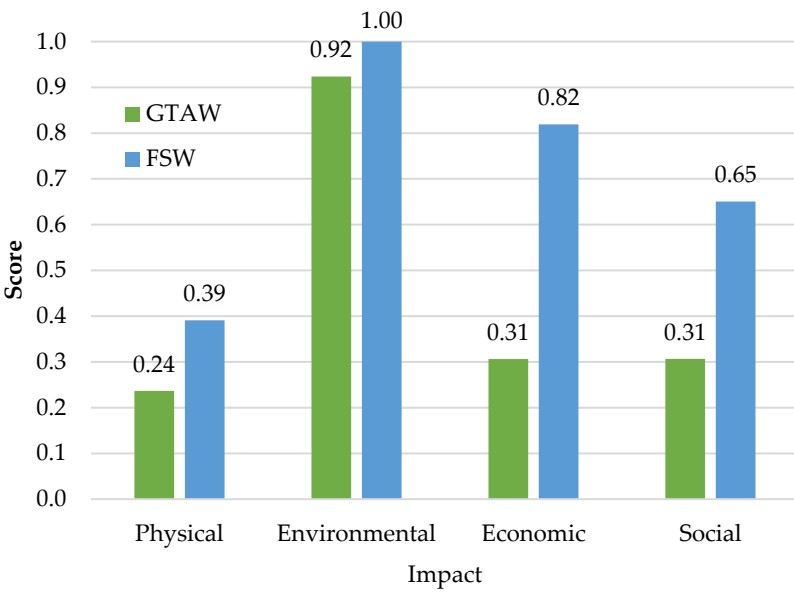

**Figure 8.** Sustainability scores for FSW and GTAW—battery trays.

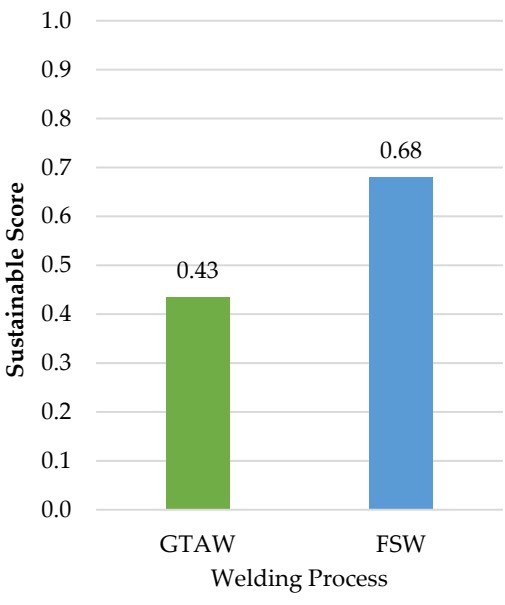

**Figure 9.** Overall score—battery trays.

## 5. Conclusions

Manufacturing industries have a growing understanding and interest in the sustainability of their products and processes, taking responsibility for people and the environment. The results of this work show the comparison of two welding processes: GTAW and FSW applied in two different cases. The methodology developed measures that were sustainable based on four different aspects: social, environmental, economic, and quality. For the Colombian case of the train component, FSW obtained the best results in all categories, with a final score of 0.78 compared to 0.69 from GTAW (11% of difference), and the major difference was obtained in the physical impact. Thus, the final mechanical properties of the weld made by conventional arc welding have lower values in terms of tensile strength and toughness. Considering the economic impact, many of the values used for the analysis are specific to the Colombian environment, given the case study proposed, so to compare it directly with the case for electric vehicle components, several modifications and considerations are required. For example, specific values related to the place of manufacturing must be considered as well as the materials that are available to be employed.

FSW has been investigated for the EV industry, and, thus, in this type of production, the final mechanical properties are considered an important factor. A sustainability score was estimated for this case in order to compare two different methods of welding (GTAW and FSW), and the final results show a higher score for FSW—0.43 compared to 0.68, a percentage difference of 36%. It is important to remark that the values mentioned previously can be higher if the case considered a higher number of pieces (more than one), and FSW tends to be better in terms of economic and environmental terms. It must also be considered that another factor to be considered is the usages of each machine. In the case of FSW, the CNC milling machines could also be used for metal removal which is not the case with GTAW as these machines have only one use.

The methodology used provides a first approximation of the use of sustainability scores as a criterion in the selection of manufacturing processes in a quantitative way, and multiple modifications can be considered to increase its potential use. For example, the weights used to assess each of the impacts considered in this analysis can be reassessed by experts in different contexts, locations, and/or industries. Furthermore, considerations about environmental impact, based only or mainly on mass changes, when evaluating autogenous welding processes are limited, and other aspects such as the energy efficiency of the process could also be incorporated. Generally, for gas-shielded welding processes, carbon footprint of argon production is mostly for transportation, and this fact could potentially negatively impact the environmental score. Also the use of argon gas could be included in the social impacts and would thus affect the health of employees.

Comparing the results obtained in the article with the results presented in the work called "A Study on Sustainability Assessment of Welding Processes", written by Jamal et al., the consumable cost has a large effect on the final results, since in the case presented in this article, the cost of the tool is lower compared with the arc welding (3.6 USD compared to 52.1 USD).

**Author Contributions:** Conceptualization, E.H. and M.C.S.; methodology, E.H. and M.C.S.; validation, E.H., M.C.S. and J.D.B.; formal analysis, E.H., M.C.S. and J.D.B.; investigation, E.H. and M.C.S.; resources, E.H. and M.C.S.; data curation, E.H. and M.C.S.; writing—original draft preparation, E.H., M.C.S. and J.D.B.; writing—review and editing, E.H., M.C.S., J.D.B. and J.M.; supervision, E.H.; project administration, E.H. All authors have read and agreed to the published version of the manuscript.

**Funding:** This research received no external funding.

**Institutional Review Board Statement:** Not applicable.

**Informed Consent Statement:** Not applicable.

**Data Availability Statement:** Data is contained within the article.

**Conflicts of Interest:** The authors declare no conflict of interest.

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
