# Peer review of "Sustainability Score Comparison of Welding Strategies for the Manufacturing of Electric Transportation Components"

_sustainability, doi:10.3390/su15118650_

Round 1

Reviewer 1 Report

Dear Authors

The authors have attempted Sustainability score comparison of welding strategies for the  manufacturing of electric transportation components.

The manuscript has been well written and adequately presented, however manuscript needs to be revision based on following comments and suggestions.

1.      In abstract , should include quantitative values of the key findings.

2.  Introduction sections, Lack of comprehensive literature support. Highlights the exciting practice and limitations of welding process of aluminum alloys.

3.      Also include more details about FSW tool details, geometry etc...

4.      Authors should justify the both welding process has two distinct process (merits and limitations), how do you select the process for comparisons?

5.      In table No 4 and 5, use uniform font size.

6.      In result and discussion, include the scientific finding and justification with relevant literature.

Author Response

We, the authors of the submitted script “Sustainability score comparison of welding strategies for the manufacturing of electric transportation components” value your detailed reviews, suggestions, and other appreciations to improve and resubmit our manuscript. We hope this new text captures all your intentions during the evaluation of the previous version and thank you for dedicating enough time for a complete reading. The notes are presented in the file attached. 

Reviewer 2 Report

This article deals with the issue of evaluating the welding process for aluminium alloys taking into account sustainability criteria. These materials are widely used in the global industry primarily because of their potential to reduce structural weight. However, they are extremely difficult to join using the welding process. This is due, among other things, to the formation of Al2O3, high thermal conductivity, high thermal expansion and a decrease in strength at the welding temperature. Due to these disadvantages, various methods have been developed, the most important of which are MIG (Metal Inert Gas) and TIG (Tungsten Inert Gas).

This article compares the popular GTAW (Gas Tungsten Arc Welding) method and the patented in 1991 FSW (Friction stir welding) method, which is also suitable for joining other materials such as titanium alloys and steels. In the FSW method, a rotary motion tool is used for heating and plasticizing the material. Owing to friction, the tool generates heat and stirs the material along the welding line. During rotation of the tool, the heated and plasticized material is moved backwards around the pin where it is mixed and press compacted. Welding takes place by mixing materials while simultaneously breaking down and evenly distributing oxides from the surface of the welded components [prof. Lacki: The application of FSW technology in aluminium structures].

However, the authors did not formulate the requirement for changing the technology of the welding processes. They did not consider the investment in tools or the cost of changing the process organisation. Table 1 compares GTAW and FSW showing only positive factors for FSW. Can processes be compared without the context of their types: job-shop, batch or continuous production?

The authors reduced the evaluation of the welding technology to a simple weighted sum of the physical, environmental, economic and social impacts. There is no discussion of the proposed formulae for calculating the impacts. Can an average be drawn from the sum of material, labour, equipment and energy costs? How is the cost of equipment understood?  Authors do not lead to a normalisation of the values used. Already at this stage, the method should be considered inappropriate for a multi-criteria assessment. Furthermore, the social impact was reduced to an ex-post prediction of the number of accidents. Not even the factors in Table 1 were used. This is important because the social impact has the second-highest assessment weight in the proposed method.

Moreover, the example calculations appear unreliable for the following reasons:

  the cost of equipment is always 0;

  no calculation of energy consumption;

  data on base material toughness and base material yield strength are missing;

  weld part cost is the same for the train component manufactured in Colombia as well as for battery trays manufactured in UK.

Additional specific comments:

1. The literature is not carefully described.

2. There is a lack of examples of calculations.

3. The abstract does not refer to the research methodology.

Author Response

(The authors gave the same response as above.)

Reviewer 3 Report

The article contributes to the field but needs substantial improvement.

1. Authors must include research questions within in the introduction section. Also add article structure at the end of the introduction section

2. Aspects of sustainability need to be covered in detail. Implications are something that will after effects of the study.

3. I have not found the theoretical background in elaborated manner. Authors must include the literature section starting with aspects of the sustainability I have few suggestions in this regards to cover the aspects of the sustainability. “Success factors for the adoption of green lean six sigma in healthcare facility: an ISM-MICMAC study” “Integrating Green Lean Six Sigma and Industry 4.0: A Conceptual Framework”, “Green Lean Six Sigma critical barriers: exploration and investigation for improved sustainable performance”.

4. Methodology section needs to be more elaborated .Currently it is feeble, add a clear depiction of the methodology.

5.  There are a few typo errors in the manuscript. Please fix the same. 

6. Conclusion must represent after effect of the study. The implications sub section needs to be added. Compare the results of your study with previous studies of the same nature.

Author Response

(The authors gave the same response as above.)

Round 2

Reviewer 3 Report

Authors have addressed all comments